# Role of the *Egr2* Promoter Antisense RNA in Modulating the Schwann Cell Chromatin Landscape

**DOI:** 10.3390/biomedicines12112594

**Published:** 2024-11-13

**Authors:** Margot Martinez Moreno, David Karambizi, Hyeyeon Hwang, Kristen Fregoso, Madison J. Michles, Eduardo Fajardo, Andras Fiser, Nikos Tapinos

**Affiliations:** 1Department of Neurosurgery, Brown University, Rhode Island Hospital, Providence, RI 02903, USA; 2Laboratory of Cancer Epigenetics and Plasticity, Brown University, Providence, RI 02903, USA; 3Department of Systems and Computational Biology, Albert Einstein College of Medicine, Bronx, NY 10461, USA

**Keywords:** promoter antisense-RNA, *EGR2*, *C-JUN*, EZH2, WDR5, AP-1, *mTOR*, genome organization, Neuregulin, ErbB2/3, YY1

## Abstract

**Background**: Schwann cells (SCs) and their plasticity contribute to the peripheral nervous system’s capacity for nerve regeneration after injury. The *Egr2/Krox20* promoter antisense RNA (Egr2-AS) recruits chromatin remodeling complexes to inhibit *Egr2* transcription following peripheral nerve injury. **Methods**: RNA-seq and ATAC-seq were performed on control cells, Lenti-GFP-transduced cells, and cells overexpressing Egr2-AS (Lenti-AS). Egr2 AS-RNA was cloned into the pLVX-DsRed-Express2-N1 lentiviral expression vector (Clontech, Mountain View, CA, USA), and the levels of AS-RNA expression were determined. Ezh2 and Wdr5 were immunoprecipitated from rat SCs and RT-qPCR was performed against AS-Egr2 RNA. ChIP followed by DNA purification columns was used to perform qPCR for relevant promoters. Hi-C, HiC-DC+, R, Bioconductor, and TOBIAS were used for significant and differential loop analysis, identifications of COREs and CORE-promotor loops, comparisons of TF activity at promoter sites, and identification of site-specific TF footprints. OnTAD was used to detect TADs, and Juicer was used to identify A/B compartments. **Results**: Here we show that a Neuregulin-ErbB2/3 signaling axis mediates binding of the Egr2-AS to YY1^Ser184^ and regulates its expression. Egr2-AS modulates the chromatin accessibility of Schwann cells and interacts with two distinct histone modification complexes. It binds to EZH2 and WDR5 and enables targeting of H3K27me3 and H3K4me3 to promoters of *Egr2* and *C-JUN,* respectively. Expression of the Egr2-AS results in reorganization of the global chromatin landscape and quantitative changes in the loop formation and contact frequency at domain boundaries exhibiting enrichment for AP-1 genes. In addition, the Egr2-AS induces changes in the hierarchical TADs and increases transcription factor binding scores on an inter-TAD loop between a super-enhancer regulatory hub and the promoter of *mTOR*. **Conclusions**: Our results show that Neuregulin-ErbB2/3-YY1 regulates the expression of Egr2-AS, which mediates remodeling of the chromatin landscape in Schwann cells.

## 1. Introduction

The peripheral nervous system (PNS) exhibits a unique ability for regeneration, primarily due to the remarkable plasticity of Schwann cells (SCs). Following nerve injury, quiescent SCs reprogram into progenitor-like repair SCs that drive the regenerative process. This involves the downregulation of promyelinating genes (such as *EGR2*/*KROX20*, *MPZ*, and *MBP*) and the upregulation of genes that define repair SCs (including *C-JUN*, *GFAP*, and *BDNF*). After axonal regeneration, these repair SCs exit the cell cycle and differentiate again into myelinating SCs [1]. A Neuregulin-ErbB2/3-Erk1/2-YY1 axis has been shown to be instrumental in regulating expression of the *Egr2* transcription factor and thus, SC myelination [2,3,4,5]. Ablation of this pathway in SCs causes severe hypomyelination and inhibition of Egr2 expression in sciatic nerves [6]. Egr2 is required for proper peripheral nerve myelination and serves as a master regulatory transcription factor in both developmental myelination and remyelination [7,8,9,10,11]. Egr2 is downregulated after axonal injury and this downregulation results in demyelination, a process known as Wallerian degeneration [11,12]. Schwann cells (SCs) possess a remarkable plasticity that allows them to reprogram and support peripheral nerve regeneration, a key aspect of neural plasticity with significant implications for therapeutic strategies targeting nerve injuries and demyelinating disorders such as Charcot–Marie–Tooth disease and multiple sclerosis [13]. Studies have shown that modulating pathways like C-JUN and Egr2 expression in SCs promote recovery post-injury, suggesting that promoting SC plasticity can facilitate axon repair and regeneration [14]. Despite these advances, the regulatory mechanisms governing chromatin remodeling and transcriptional silencing in SCs remain largely unexplored.

We have previously mapped an antisense RNA complementary to the *EGR2* promoter in SCs and showed that its expression increases following peripheral nerve injury and it mediates the recruitment of EZH2 and H3K27me3 on the *EGR2* promoter to induce transcriptional silencing of *EGR2* [12]. However, the regulation of Egr2-AS expression by the signaling network that controls SC myelination and nerve injury response, as well as the role of the Egr2-AS in regulating the Schwann cell chromatin landscape, has not been established yet. In recent years, non-coding RNAs (ncRNAs) have emerged as major regulators of the global chromatin landscape. ncRNAs are RNA molecules that are not translated into proteins and are implicated in numerous cellular processes including transcription, mRNA splicing, and protein translation [15,16,17]. ncRNAs impact spatial genome organization by modulating perinuclear chromosome tethering, the formation of major nuclear compartments, chromatin looping, and various chromosomal structures [18]. The roles of ncRNAs often intersect with various other protein or nucleic acid components of the genome structure and function. Although the role of ncRNAs as modulators of the 3D genome organization of the X chromosome, various developmental genes, and repetitive DNA loci [18] has been described, the exact mechanistic roles of the low-copy promoter-associated antisense RNAs (AS-RNAs) in gene regulation, chromatin remodeling, and spatial genome organization are not known.

Here we show that the Neuregulin-ErbB2/3-YY1^Ser184^ axis regulates expression of the Egr2-AS, which functions as a molecular scaffold to bring WDR5 and EZH2 together with activating (H3K4me3) and repressing (H3K27me3) histone markers on the promoters of *C-JUN* and *EGR2*, respectively. Expression of the Egr2-AS increases chromatin accessibility in promoters with footprints of the AP1 transcription factor family. Finally, interrogation of the 3D genome architecture using Hi-C reveals that expression of the Egr2-AS in SCs induces reorganization of the 3D genome architecture, mild changes in the hierarchical topologically associating domains (TADs), and increased transcriptional regulation on an inter-TAD interaction between a super-enhancer regulatory hub and the promoter of *mTOR*. We propose that the Egr2-AS modulates the SC chromatin landscape under the regulatory control of a Neuregulin-ErbB2/3-YY1^Ser184^ axis. By examining how Egr2-AS drives epigenetic modifications and 3D genome reorganization, our research provides new insights into the molecular mechanisms that promote SC plasticity and regeneration, offering potential novel therapeutic targets for peripheral nerve damage and related neuropathies [19].

## 2. Materials and Methods

Methods for cell cultures, transfections, qPCR, TF activity assays, and primer sequences are included in the Appendix A.

### 2.1. RNA-seq

For each biological replicate, total RNA was isolated from the control cells, Lenti-GFP-transduced cells, and cells overexpressing Egr2-AS (Lenti-AS) using Trizol and the PureLink RNA Mini Kit, according to the manufacturer’s protocol, followed by DNase treatment. Two micrograms of RNA were submitted to GENEWIZ for library generation and sequencing on the Illumina HiSeq 2500 using a 50 bp paired-end run. Sequencing reads were aligned to the rn6 genome assembly with hisat2 [18]. Gene-wise read summarization was performed with featureCounts using Refseq exon coordinates, as specified in the rn6 refGene file available in the UCSC Genome Browser database [20,21]. Differential gene expression analysis was performed in R via the DEBrowser DESeq2 platform [22,23]. We used an adjusted *p*-value cut-off of 0.05 and a fold change of 1.3.

### 2.2. ATAC-seq

For each biological replicate, 100,000 cells were used to prepare ATAC-seq libraries following a modified protocol from Kaestner’s lab [24]. After the transposition reaction using the Nextera DNA Library Prep Kit (San Diego, CA, USA), DNA was isolated with a MinElute Reaction Cleanup Kit (Hilden, Germany). To create the libraries, we used the NEBNext High-Fidelity 2X PCR Master Mix. The primers (universal and indexing) used are listed in Appendix A. For multiplexing, we used the same universal primers for every sample, and each sample had its unique indexing primer. Quality control (QC) of the libraries was performed before sequencing using the KAPA Library Quantification Kit from Illumina Platforms. Libraries were sequenced by GENEWIZ using the Illumina HiSeq 2500 system to acquire 150 bp paired-end sequence reads, reaching 200 million genomic reads per sample to detect open and closed chromatin regions and identify transcription factor binding sites through transcription factor footprinting. QC, showing an equal distribution of library fragments between Lenti-GFP and Lenti-AS, was initially performed (Appendix A). Sequencing reads were aligned to the rn6 genome assembly with bowtie2, and duplicate reads were removed with Picard tools [25,26]. Promoters were defined as 2000 bp regions positioned between −1500 to +500 nucleotides with respect to the transcription start site, as annotated in Refseq. Peaks of chromatin accessibility were detected using MACS2, and those located at promoter regions were identified using the intersectBed command in BEDtools (previously “intersectBed routine of Bedtools”) [27,28]. To compare the relative promoter accessibility, ATAC reads mapping promoters were quantified with featureCounts and the promoter locations defined above. Differential promoter accessibility was determined with the limma package [29]. A *p*-adjusted value of 0.05 and fold change of 1.5 were used as the cut-offs. Motif binding sites with differential activity were analyzed through EnrichR to identify enriched gene ontology categories [30].

### 2.3. Lentiviral Expression of the AS-RNA

The sequence of the Egr2 AS-RNA identified by 5′-RACE was cloned into the pLVX-DsRed-Express2-N1 lentiviral expression vector (Clontech, Montain View, CA, USA). The sequence and details for cloning have been published [12]. As a control, we used the plasmid pLVX-AcGFP1-N1 from Clontech. After transformation of the competent *E. coli* strain Stbl3 (Life Technologies, Carlsbad, CA, USA) and further purification with the endotoxin-free MaxiPrep Kit (QIAGEN, Venlo, The Netherlands), we produced the lentivirus with the Lenti-X HTX Packaging System from Clontech.

To determine the levels of AS-RNA expression following lentiviral infection, we isolated RNA from 1/10th of the transduced cells using Trizol and the PureLink RNA Mini Kit, according to the manufacturer’s protocol, following DNase treatment. A sample of 300 ng of RNA was reverse-transcribed to cDNA using the SuperScript III Strand Synthesis System. For all qPCRs reported in the paper, we performed a no reverse transcription (RT) control amplification to verify the absence of genomic DNA contamination with GAPDH primers. We ran a qPCR using SYBR Green PCR Master Mix with the AS-RNA-specific primers (listed in Appendix A) at a concentration of 250 nM. Relative mRNA levels were normalized to GADPH and quantified using the comparative Ct method, and fold change was calculated compared to each control.

### 2.4. RNA Immunoprecipitation (RIP)

Ezh2 and Wdr5 were immunoprecipitated from rat Schwann cells (RSCs) using the RNA ChIP-IT^®^ magnetic chromatin immunoprecipitation kit (Active Motif). This kit collects RNA that is bound to chromatin or to proteins that are bound to chromatin. Briefly, cells were washed with ice-cold RNase-free PBS, following fixation with 1% formaldehyde in PBS to ensure crosslinking of RNA with chromatin-bound proteins Crosslinking was stopped by adding 1/10th of the volume of 1 M Glycine. After three more washes with ice-cold RNAse-free PBS, cells were collected in a 1.5 m tube and lysed with the lysis buffer provided by the kit. After centrifugation, chromatin was sheared to 100–1000 bp using the Qsonica Q800R3 sonicator DNA and chromatin shearing system (a 1/10 aliquot was taken per sample to ensure consistent sonication by isolating DNA, then analyzed in an Agilent fragment analyzer). Sheared chromatin was quantified and then subjected to DNase treatment before proceeding to immunoprecipitation with the magnetic beads. For each immunoprecipitation, equal amounts of sheared chromatin (10 μg) and 1 μg of ChIP-grade antibody (list of antibodies in Appendix A) were used. For each sample, we separated 10 μL of chromatin and kept it at −80 °C to be used as the input sample. An isotype antibody control (anti-rabbit IgG) was used per sample to account for unspecific binding to the beads and the proteins. We incubated every tube in an end-to-end rotator overnight at 4 °C. After 6 washes in kit-specific washing buffers, we eluted the RNA using a kit-specific elution buffer, and the eluate was incubated in Proteinase K. In this step, we recovered the input samples and the subsequent steps were performed together with the immunoprecipitated samples. After degrading the proteins and reversing the crosslinks, we performed RNA purification with Trizol. After an extra incubation of DNase I, we performed RT-qPCR against AS-Egr2 RNA on all the samples. The results were analyzed following the manufacturer’s recommendations. Input normalized values were calculated as the percentage of fold enrichment (22DDCt) of the RNA recovered after the immunoprecipitation compared to the RNA input. Then, we deducted the IgG values to account for non-specific binding; then, to show differences between the Lenti-AS-Egr2 RNA overexpression and the Lenti-GFP control while correcting the variability between different experiments, we calculated the fold change, defined as the ratio of the changes between the amount of immunoprecipitated RNA after the AS-RNA overexpression compared to control.

### 2.5. Chromatin Immunoprecipitation (ChIP)

For the DNA chromatin immunoprecipitation (ChIP), we used the ChIP-IT High Sensitivity^®^ kit from Active Motif. Briefly, cells in culture were trypsinized from the plates and snap-frozen at −80 °C to test for AS-Egr2 RNA overexpression and GapmeR inhibition (Appendix A). Cells were washed with ice-cold PBS following fixation with a complete cell fixative solution. Crosslinking was stopped by adding 1/20 of the volume of stop solution. After 3 more washes with ice-cold PBS, cells were collected in a 1.5 mL tube and resuspended in 5 mL of Chromatin Cell Prep Buffer and lysed with a Dounce homogenizer on ice. After centrifugation, the pellet was resuspended in the ChIP Buffer. Chromatin was sheared to 100–1000 bp using the Qsonica Q800R3 sonicator DNA and chromatin shearing system (a 1/10 aliquot was taken per sample to ensure consistent sonication by isolating DNA, then analyzed in a fragment analyzer). For each immunoprecipitation, equal amounts of sheared chromatin (10 μg) and 4 μg of ChIP-grade antibody (list of antibodies in Appendix A) were used. For each sample, we separated 10 μL of chromatin and kept it at −80 °C to be used as an input sample. An isotype antibody control (rabbit IgG) was used per sample to account for non-specific binding. We incubated every tube using an end-to-end rotator overnight at 4 °C. Protein G agarose beads were added to each immunoprecipitation and tubes were incubated at 4 °C for 3 h with rotation. Beads were then washed using the ChIP filtration columns provided by the kit, and five subsequent washes were performed with a wash buffer. DNA was eluted using a kit-specific elution buffer, and the eluate was incubated in Proteinase K. In this step, we recovered the input sample, and the subsequent steps were performed together with the immunoprecipitated samples. After this incubation, DNA purification columns provided by the kit were used to purify the DNA. qPCR was performed for the appropriate promoters listed in Appendix A. Input normalized values were calculated as fold enrichment (2DDCt), then we deducted the control IgG values to account for non-specific binding.

### 2.6. Hi-C Capture Method

Hi-C was performed using the Arima-HiC kit. As per their protocol, we tested the amount of input sample beforehand and used 2 million cells per each independent sample. We performed QC on the samples in every recommended step. Sonication was performed using the Covaris s220 instrument to a fragment size of 300–400 bp (settings: power: 5, duty factor: 10%, cycles per burst: 200, and time: 60 s per process). We used the KAPA Hyper Prep Kit to generate libraries. The primers used for indexing were obtained from Illumina, the libraries were quantified, and QC was performed using the KAPA Library Quantification Kit. Libraries were sequenced by GENEWIZ using the Illumina HiSeq 2500 system to acquire 150 bp paired-end sequence reads, reaching 300 M reads for each sample based on the total usable reads >20 Kb.

QC for HiC replicates was performed (Appendix A), and raw and ICE-normalized contact matrices were prepared with HiC-Pro [31,32]. ICE-normalized and condition-merged HiC-Pro matrices were converted to Juicer .hic files for the resolutions 10 Kb and 100 Kb using a wrapper for the Juicer tools pre-command in the HiCDPlus Bioconductor package version 1.0.0 [33,34].

### 2.7. Significant and Differential Loop Analysis

Significant loops (q-value ≤ 0.05) and differential loops (log2FC cut-off = 1, *p*-value < 0.05) across conditions were identified using HiCDCPlus version 1.0.0 with raw HiC-Pro matrices and bed files of two AS replicates and two GFP replicates at 10 Kb resolution [35]. To filter biases in the denser genomic regions in close proximity and noise in the sparser regions that are farther apart, only loops between 50 Kb and 2 Mb were called significant.

We categorized the loops into three sets: (1) static for significant loops that did not meet the differential loops threshold, (2) dynamic gain for significant loops that were differential with a log2FC > 1, and (3) dynamic lost for significant loops that were differential with a log2FC < −1. Dynamic loops represent interactions that, after antisense overexpression, were gained or lost relative to their interaction frequencies in the GFP control samples.

### 2.8. Identification of Clusters of cis-Regulatory Elements (COREs)

The R package CREAM version 1.1.1 was used to identify clusters of cis-regulatory elements, or COREs [19]. The bed files for the peaks were used as an input and the program was run with the parameters MinLength = 1000 and peakNumMin = 2 for all the Lenti-AS and Lenti-GFP samples.

### 2.9. Annotation/Identification of CORE–Promoter Loops

Gene symbols obtained from the RNA-seq analysis were first mapped to stable rn6 Entrez Gene identifiers, then to transcripts, and finally to 2200 bp-width promoters using the Bioconductor packages TxDb.Rnorvegicus.UCSC.rn6.refGene, BSgenome.Rnorvegicus.UCSC.rn6, and org.Rn.eg.db [31,32,36]. We defined CORE–promoter loops as loops that consisted of one anchor intersecting with at least one promoter and the other anchor intersecting with at least one CORE. Anchors of CORE–promoter loops had at least a 1 bp-overlap with the promoters or COREs. A subset of these loops had complete overlaps at anchors in which an entire promoter or CORE was contained in the 10 Kb loop anchor or, for COREs greater than 10 Kb in width, the entire loop anchor was contained in the CORE.

### 2.10. Genome Wide Differential TF Activity at Promoter Sites

Peaks were annotated against the rn6 genome. The R package diffbind from Bioconductor was used to compare the similarity of the peak read counts via correlation heatmaps and principal components analysis [37]. The R package DESeq2 from Bioconductor was used to detect differential peak regions [23]. To do so, a consensus set of peaks was first generated via the runConsensusRegions function in the *soGGi* package [38], then read counts of each peak range in the consensus peak set were calculated via the featureCounts function in the Rsubread package for each sample [21,39,40]. The resulting counts matrix was then processed via DESeq, and differentially accessible chromatin regions between the control and experimental samples were defined as peak regions with an absolute fold change greater than 1.5 and a *p*-value less than 0.05. Genes associated with differentially accessible peaks were identified through a many-to-many mapping algorithm implemented in the seq2gene function from the ChIPseeker package [41]. Promoters with differential accessibility were defined as differentially accessible peak regions within 3000 bp upstream and downstream of a TSS in the rn6 genome. DAStk was used to quantify significant differences in the transcription factor activity at differentially accessible promoter sites [42].

### 2.11. Site-Specific TF Footprint/Binding Predictions

TOBIAS was used to identify specific footprint locations at DNA regulatory elements [43]. First, the ATACorrect function was run to correct for tn5 insertion bias. Next, depletion signals (negative counts) and general accessibility at surrounding regions were used to derive a binding score. Finally, to make TF binding activity predictions, the footprint scores were matched to binding motifs from the 2020 JASPAR CORE database [44]. TF binding activity was restricted to accessible loop anchor sites using genomic contact regions derived from the Hi-C data. Finally, the exact sites predicted to be bound were extracted for further site-specific downstream analysis.

### 2.12. Site-Specific TF Binding Predictions at Regulatory Elements

All intersections of TF footprints with anchors, promoters, and COREs had an overlap of at least 1 bp. We identified TF footprints that intersected with COREs and with all anchors overlapping with COREs. TF footprints that intersected with each of the three loop anchors interacting with the mTOR promoter (i.e., anchors containing COREs) were grouped into families and classes based on the 2020 and 2022 JASPAR CORE databases [44,45]. To identify significant differences between the AS and GFP footprint scores (binding scores) at each anchor, we performed Wilcoxon rank sum tests in R on (1) the full list of TF footprints that intersected with the loop anchor and were present in both the AS and GFP samples and (2) the same list grouped by TF class.

### 2.13. Identification of TADs

The C++ software OnTAD version 1.4 was used to detect hierarchical TADs in ICE-normalized and condition-merged .hic files at a 10 Kb resolution [46]. The OnTAD executable file was compiled under gcc 8.3 and run with default parameters.

### 2.14. Intra and Inter TAD Loops Localization

We defined intra-TAD loops as loops with both anchors intersecting with the same outermost TAD and inter-TAD loops as loops with anchors intersecting with different outermost TADs. All intra- and inter-TAD loop anchors were annotated with genes with promoters that overlapped with any region of the anchor.

### 2.15. Identification of A/B Compartments

The eigenvector command in Juicer tools version 1.19.02 was used to call A/B compartments in ICE-normalized and condition-merged .hic files at a 100 Kb resolution [33]. The signs of the output eigenvectors indicate the compartment, but the signs are arbitrary in each set of eigenvectors calculated for each chromosome and condition. Therefore, for the regions of interest, eigenvectors were plotted alongside ATAC-seq narrow peaks using the plotgardener R package to determine whether they corresponded to compartment A or B [47].

### 2.16. Software & Data Availability

Hi-C analysis scripts are available at https://github.com/hyeyeon-hwang/HiC-Analysis-of-Egr2-AS-RNA accessed on 14 April 2022. All datasets (RNA-seq, ATAC-seq, Hi-C) have been deposited to the GEO Omnibus and can be accessed through: https://www.ncbi.nlm.nih.gov/geo/query/acc.cgi?acc=GSE201627 accessed on 16 April 2023.

### 2.17. Plasmid Availability

The lentivirus construct (Lenti-AS) was deposited to Addgene and is available to order (#177737).

## 3. Results

### 3.1. Neuregulin-ErbB2/3 Regulate YY1 Phospho-Ser184 Expression and Binding of YY1 to the Egr2-AS

Recently, we showed that a phospho-switch between YY1-Ser184 and YY1 controls binding of the Egr2-AS to YY1 and regulates its transcriptional effects [12]. To determine if sciatic nerve injury induces dephosphorylation of the serine residues of YY1, we performed immunoprecipitation (IP) of YY1 using a verified YY1 antibody followed by Western blotting with total phosphoserine antibody and total YY1 as a loading control. This showed that 12 h after sciatic nerve injury, YY1 is dephosphorylated on Serine residues (Figure 1A). To determine whether the dephosphorylation of YY1 is specific to Serine184, we developed an affinity purified pSer184-specific antibody by immunizing rabbits with the phosphopeptide KSGKK(pSer)YLGGGAGAC, which corresponds to the sequence spanning Ser184 of YY1. For the control, we generated an antibody against the non-phospho YY1 peptide KSGKKSYLGGGAGAC. After 12 h following sciatic nerve injury, we observed a significant reduction in specific pSer184 on YY1 compared to non-injured nerves using a quantitative phospho-YY1^Ser184^ ELISA (Figure 1B). YY1 is the link between Neuregulin-ErbB2/3 signaling and the transcriptional regulation of SC myelination [5]. To determine whether the pharmacologic inhibition of the Neuregulin-ErbB2/3 axis with PKI 166 (PKI) (Appendix A) inhibits YY1-Ser184 phosphorylation, we performed a quantitative phospho-YY1^Ser184^ ELISA in SCs treated with or without PKI 166. This showed significant inhibition of YY1-Ser184 phosphorylation (Figure 1C). Finally, to determine whether the inhibition of Erb2/3 signaling with PKI 166 results in increased association of the Egr2-AS with YY1, we performed RNA immunoprecipitation (RIP) with a pan-YY1 antibody and detected the Egr2-AS with and without addition of PKI 166. We thus showed that inhibition of Neuregulin-Erb2/3 with PKI 166 results in a significant increase in binding of YY1 to the Egr2-AS (Figure 1D).

### 3.2. Expression of the Egr2-AS Results in Transcriptomic Changes in Schwann Cells

To determine the effect of the Egr2-AS on global gene expression, we infected SCs with a lentivirus expressing the Egr2-AS (Lenti-AS), or Lenti-GFP as control as previously described [12], and performed RNA-seq. We noted 561 differentially expressed genes. A total of 111 were downregulated and 450 were upregulated (Figure 1E). Upregulated genes were enriched for mTOR signaling and cell cycling biological processes. Downregulated genes were enriched for ERK1/ERK2 signaling, ErbB signaling pathways, and MAPK regulation (Figure 1F,G).

### 3.3. The Egr2-AS Inhibits EGR2 and Activates C-JUN Expression in SCs

Since EGR2 and C-JUN transcription factors control the balance of differentiation and dedifferentiation of SCs [48], we examined whether overexpression of the Egr2-AS has opposing effects on the expression of EGR2 and C-JUN. We infected SCs with Lenti-AS or Lenti-GFP and showed that ectopic expression of the Egr2-AS inhibits the expression of the EGR2 protein and induces the expression of C-JUN (Appendix A). To rule out the possibility that the increase in expression of C-JUN is an indirect effect of the inhibition of EGR2 by the Egr2-AS, we inhibited EGR2 expression in SCs independent of the Egr2-AS by using EGR2 targeting siRNAs to show that although EGR2 protein expression is significantly inhibited by the EGR2 targeting siRNAs, C-JUN expression is not affected (Appendix A). Taken together, our results suggest that the Egr2-AS could regulate C-JUN through a different molecular mechanism that does not depend on the direct inhibition of EGR2.

### 3.4. The Egr2-AS Interacts with EZH2 and WDR5 to Enable Targeting of H3K27me3 and H3K4me3 at Egr2 and C-JUN Promoters

To investigate the regulatory role of Egr2-AS on Egr2 and C-JUN through chromatin-modifying enzymes (CMEs) and histone marks, we conducted RNA immunoprecipitation (RIP) and chromatin immunoprecipitation (ChIP) in Schwann cells (SCs) following the overexpression of Egr2-AS. In the RIP experiments, we show a greater pull-down of Egr2-AS with antibodies against Ezh2 and Wdr5 (Figure 2A). This indicates that Egr2-AS preferentially interacts with these chromatin-modifying proteins following its overexpression. However, this interaction is not solely due to the overexpression itself, as the total amount of the Egr2-AS RNA is normalized based on the input sample. The ChIP assays demonstrated a significant increase in the binding of the repressive histone mark H3K27me3 to the Egr2 promoter (Figure 2B, left) and the activating mark H3K4me3 to the C-JUN promoter (Figure 2B, right) after Egr2-AS overexpression. These interactions were reverted by specific inhibitory oligonucleotides (GapMers) targeting the Egr2-AS [12], confirming that the observed histone modifications at the *EGR2* and *C-JUN* promoters are dependent on the instructive role of Egr2-AS. This suggests that Egr2-AS orchestrates the regulatory landscape for Egr2 and C-JUN through its interactions with EZH2 and WDR5, which methylate H3K27 and H3K4 in the *EGR2* and *C-JUN* promoters, respectively, thereby regulating their expression in injury response [1].

### 3.5. The Egr2-AS Induces Chromatin Remodeling and Increased Binding of the AP-1/JUN Family of TFs

We performed ATAC-seq to capture genome-wide epigenetic profile changes induced by the Egr2-AS. First, we performed a PCA analysis which showed that control and treated samples closely cluster together (Figure 3A and Appendix A), suggesting that the epigenetic changes induced by the AS-RNA are reproducible. We then performed a differential accessibility analysis at the promoters and observed that the Egr2-AS increased accessibility at 109 sites and decreased it at 467 sites (Figure 3B). Genes with increased accessibility showed significant enrichment for the AP-1 transcription factor network (Figure 3C). Promoters of the genes of the AP-1 pathway (such as *JUN*, *JUNB*, *JUND*, *FOSB*, and *FOS*) saw a significant increase in accessibility following Egr2-AS overexpression (Figure 3D and Appendix A). Moreover, we showed that the Egr2-AS overexpression significantly increased the TF activity of C-JUN (Appendix A). The increase in C-JUN transcriptional activity was not secondary to the effect of the AS-RNA on EGR2, since we are unable to replicate this finding when we treat SCs with EGR2-specific siRNA. ATAC-seq analysis also revealed a significant reduction in chromatin accessibility at the Egr2 gene locus after the Egr2-AS overexpression (logFC = −1.21, adj.*p*.Val = 3.82 × 10^−6^), indicating that the chromatin is approximately 2.3 times less accessible compared to the control condition. Moreover, the Egr2 promoter activity, as shown in a luciferase assay, was inhibited by the Egr2-AS overexpression (Appendix A). These results highlight the dual function of Egr2-AS by its ability to both enhance chromatin accessibility and the transcriptional activity of AP-1 pathway genes, including C-JUN, while simultaneously reducing accessibility and repressing activity at the Egr2 gene locus, indicating its selective regulatory role in gene expression.

### 3.6. Expression of the Egr2-AS Results in Genome Reorganization and a Gain in Stable Loops Associated With Genes Enriched for AP-1 TF Network and NOTCH 1 Signaling

Since the Egr2-AS can interact with both activating and repressive histone-modifying enzymes, we hypothesized that expression of the Egr2-AS may induce spatial genome reorganization in SCs to epigenetically affect global transcriptional programs. We performed Hi-C following expression of the Egr2-AS using Lenti-AS and compared the genome architecture with SCs expressing Lenti-GFP as control in duplicates (n = 2). Overexpression of the Egr2-AS resulted in a decrease in total loops (Appendix A). Lenti-GFP samples had an average detection of 86,382 significant interactions (loops), while Lenti-AS samples had 68,901 loops on average. Loops in the Egr2-AS samples made shorter range interactions compared to the control samples (Appendix A). We identified the stable loop structures defining contact points between the genomic loci pairs (loop anchors) from the contact frequency map (Hi-C map) (Appendix A). Next, we performed differential loop analysis to identify changes in the putative loop interactions that were due to the Egr2-AS overexpression. These were labelled as static, gained, or lost loops according to the changes in interaction frequency. Though most loops/interacting loci were static, we found 545 loops that changed following Egr2-AS overexpression (Figure 4A). A total of 104 loops were gained and 441 lost. The dynamic loops made shorter range interactions than static loops (Figure 4B). Overall, most of the dynamic loops lost interactions, and this was consistent across all chromosomes (Appendix A).

We next proceeded to annotate the gene promoters existing at the loop anchors (Figure 4C). We found that promoters annotated to the gained loops were enriched for the AP-1 TF network and NOTCH1 signaling (Figure 4D-left panels). Lost loops were enriched for C-MYC signaling, β-CATENIN, and SMAD2/3 signaling (Figure 4D, right panels).

### 3.7. Integrated 3D Genome Reconstruction Reveals Clusters of cis-Regulatory Elements (CORES) and Associated Promoters

Previous work has shown that DNA regulatory regions such as promoters, enhancers, silencers, and insulators are in open chromatin, nucleosome-depleted areas [49]. A limited set of such accessible sites is composed of large intergenic clusters of accessible chromatin sites (<20 Kb apart) that are cell type-specific and hence, reminiscent of super-enhancers [50,51,52]. These regions were dubbed “clusters of cis-regulatory elements” or COREs. We identified an average of 309 and 472 of such clusters (CORES) in the control and AS-RNA samples, respectively (Appendix A). Additionally, we noted a TF occupancy at 40% of the COREs, with a median occupancy of 155 footprints (Figure 5A and Appendix A).

Distal DNA regulatory elements with significant TF footprints are known to form regulatory, stable loops at promoters. To identify these loop structures, we first identified loops whose anchors join a distal CORE and its target promoter(s). Approximately 10% (157/1563 total CORES across all samples) contacted at least one promoter site (Figure 5B). There were, in total, 192 long-distance genomic interactions between the CORES and promoters. These interactions occurred at median genomic distances of 365 Kb and 340 Kb in the GFP control and AS-RNA, respectively (Appendix A). From the overlap analysis of differential HiC loop anchors (10 Kb) and COREs, we found that multiple loop anchors overlapped with multiple COREs. In total, 34 unique differential HiC loops overlapped with at least one CORE. Of the 104 gain loops, ten unique gain loops had at least one anchor that overlapped with at least 1 bp of at least one CORE. Of the 441 lost loops, 24 unique lost loops had at least one anchor that overlapped with at least 1 bp of at least one CORE. The set of unique COREs that overlapped with the differential loop anchors included 19 and 16 COREs from the Lenti-AS and Lenti-GFP samples, respectively. Pathway enrichment analysis of the CORES interacting genes showed top enrichment for translation, mTOR signaling, protein metabolism, PI3KC1/AKT signaling, and AMPK signaling (Figure 5C).

### 3.8. Expression of the Egr2-AS Induces Regulatory Changes Between the mTOR Promoter and Its cis-Regulatory Elements

Of the 192 CORE–promoter interactions, we focused on regulation at the mTOR promoter. Beyond its established [53] significance in SC plasticity, we elected to further probe the regulation of the mTOR promoter since it interacts with a large cluster of regulatory elements (a region covering ~100 Kb), reminiscent of a super-enhancer (Figure 5D). We tested whether this regulatory region uniquely interacts with the mTOR region by screening for all other possible interactions and found none. This suggests that we discovered an mTOR regulating super-enhancer/regulatory region located ~336 Kb bases away from the mTOR promoter.

Lower resolution inspection of contact frequencies at the COREs–mTOR locus (500 Kb–~1 Mb resolution) revealed that mTOR and its regulating super-enhancer region are in different domains (Appendix A). Closer inspection of the COREs–mTOR promoter interaction showed three main points of contact that originate from the COREs region and perfectly contact the mTOR promoter (Appendix A). We further differentiated these contact points based on the distance from the nearest TAD boundary, labelled “nearest distance”, “mid distance”, and “farthest distance” to the TAD boundary (Figure 5D and Appendix A). These contact points harbored 408 TF footprints. We found a significant increase in the TF binding score at the nearest and mid-distance contact points of the TAD boundary sites following Egr2-AS expression (Figure 6A). Specifically, expression of the AS-RNA induces a significant increase in the TF binding score of the AP-1 family, FOX family, Tryptophan cluster class, TEA domain class, and RHR class nearest to the TAD boundary sites, and a significant increase in the binding scores of the GLI family and C2H2 TFs at the mid-distance contact point of the TAD boundary sites (Figure 6B). Next, we clustered the TF footprints present at the nearest distance to the TAD boundary of the mTOR promoter and its COREs in TF communities based on the presence of families of TF footprints. The nearest to TAD contact points had 342 TF footprints, with a predominance for the AP-1 family of TFs, structural TFs such as CTCF, VEZF1, and MAZ, and for zinc fingers of the Kruppel family (Appendix A).

### 3.9. The Egr2-AS Induces Changes at the mTOR Interdomain Regulatory Hub

The mTOR and COREs harboring TADs, which both occupy active compartments (A compartments), exhibit preferential interdomain interactions (Appendix A). While most of the interactions occur within TADs, 30% of interactions occur between TADs. Additionally, we found that these interdomain interactions occurred mostly in non-adjacent TADs (Appendix A). Next, we probed for changes occurring at the mTOR regulatory hub following Egr2-AS expression. We noted the formation of a new boundary at the 3′ end of the mTOR gene after Egr2-AS expression (Figure 6C). This new boundary resulted in the formation of a nested sub-structure whereby mTOR is more insulated from other possible intra- and inter-TAD interactions. The average interaction frequencies in the left and right sub-TAD that the new TAD boundary created were 1.73 and 1.98, respectively. The OnTAD confidence scores for the left and right sub-TADs were 0.766 and 0.631, respectively. OnTAD confidence scores that were positive were considered true TADs. Such changes may function to maximize mTOR–CORE interaction while minimizing other interactions. The structural changes at the mTOR interaction hub were accompanied with moderate but statistically significant increases in mTOR transcript expression levels following Egr2-AS expression (Figure 6D).

## 4. Discussion

Recently, we demonstrated that an *EGR2* promoter antisense RNA (Egr2-AS) is induced following in vivo sciatic nerve injury. The Egr2-AS inhibits *EGR2* transcription via the recruitment of components of the PRC2 complex (EZH2, H3K27me3) to the *EGR2* promoter, while inhibition of the Egr2-AS expression delays demyelination in a sciatic nerve transection model [12]. It has been shown that long non-coding RNAs can have opposing regulatory effects by simultaneously binding to epigenetic activators and inhibitors [54,55], but this role has not been established for promoter antisense RNAs. We show here that the Egr2-AS recruits activating and repressing histone marks on the promoters of *C-JUN* and *EGR2*, respectively, and increases chromatin accessibility in promoters that exhibit increased footprints of the AP-1 family TFs. Since expression of the Egr2-AS induces chromatin remodeling, we questioned the possibility that its expression could affect spatial genome organization which could result in global transcriptional regulation from a single epigenetic input.

Recently, it was shown that subtle changes in chromatin loop contact can lead to significant changes in gene expression [56]. These loops can bring together regulatory elements with their target sequences, which can be distant from one another. After the Egr2-AS expression, we observed a decrease in the total number of loops formed, and these loops made shorter range interactions relative to the control samples. It has been suggested that larger loops may have a structural role and could therefore be involved in the formation of insulated neighborhoods to protect and stabilize interactions between promoters and regulatory elements, while shorter range interaction loops may be more regulatory [35,57,58]. In addition, our Hi-C data show enrichment of genes involved in AP-1 and NOTCH1 signaling at the gained loop anchors, suggesting that AP-1 and NOTCH1 signaling networks experience a change in regulation due to the increased contact with distal regulatory elements following expression of the Egr2-AS.

Clusters of COREs are reminiscent of super-enhancers, which are large clusters of active enhancers that drive gene expression and confer specific cell identity [59]. We identified COREs with high TF activity, suggesting that these are hubs for transcriptional regulation. These CORES form long-range interactions with promoters of genes that belong to the mTOR, AKT, AMPK, and protein translation pathways. Areas of the genome where chromatin interactions are more frequent are termed TADs [46,60]. The positions of TADs within the genome are stable between several cell types and even across species [59,60], suggesting that TADs are architectural units that house regulatory interactions. Recent work has shown the existence of a certain hierarchy with TADs, where domains are included within other domains (meta-TADs) through TAD–TAD interactions [59,60], and this level of organization correlates with cell-specific transcriptional and epigenetic regulation. However, the molecular mechanisms that regulate this cell-specific architecture are not known. In addition, the possibility that changes in TAD–TAD interaction frequencies could have profound effects on cell identity programs opens exciting possibilities for understanding and even modifying cell-specific responses. To gain insight into the potential role of the Egr2-AS in modulating interactions between different TADs, we focused on the CORE–promoter interaction of *mTOR* since this CORE has all the attributes of a regulatory hub.

Analysis of our Hi-C data following expression of the Egr2-AS showed that the promoter of *mTOR* and its regulatory COREs are positioned in separate TADs and form three distinct loop contacts. Expression of the Egr2-AS significantly increases the TF binding scores of AP-1, FOX, Tryptophan class, RHR class, and GLI families that form TF communities on the inter-TAD contact boundaries between the *mTOR* promoter and its COREs. In addition, expression of the Egr2-AS induces the formation of a new boundary at the *mTOR* gene that insulates *mTOR* from other intra- or inter-TAD interactions. The role of *mTOR* on SCs varies depending on the cellular state; it is crucial for myelination/differentiation of SCs during development [61,62], but in adulthood mTOR has been proposed to contribute to SCs’ remarkable plasticity [53]. It has also been described as one of the first proteins that is upregulated after nerve injury, promoting C-JUN elevation and SC dedifferentiation [63].

We propose here that the Egr2-AS could function as a molecular recruiter of histone-modifying enzymes and may regulate interdomain regulatory hubs that are cell-specific. Our results raise some intriguing possibilities and interesting mechanistic questions. Are these meta-TAD interactions formed by chromatin folding on single cells, or are they results of aggregate populations of cells? Single-allele chromatin interactions do reveal regulatory hubs [64,65], supporting the notion that these interactions occur in individual cells. What directs the binding of Egr2-AS–histone mark complexes on specific TADs housing regulatory hubs? The possibility of TF feedback regulation of the Egr2-AS should be explored, as it could explain the cell-specific autonomous regulation. Finally, the ability of the Egr2-AS to regulate spatial genome architecture in SCs opens exciting possibilities for the development of RNA therapeutics targeting promoter AS-RNAs for nerve regeneration.

In conclusion, our findings elucidate the crucial role of *EGR2* promoter antisense RNA (Egr2-AS) in modulating chromatin architecture and transcriptional regulation in Schwann cells following nerve injury. The ability of Egr2-AS to recruit both activating and repressive histone marks to opposite gene regulators (*EGR2* and *C-JUN*), coupled with its influence on chromatin looping and spatial genome organization, highlights its contribution to the regulatory network involved in peripheral nerve regeneration. The observed alterations in inter-TAD interactions and the insulation of critical genes, such as mTOR, further underscore the significance of Egr2-AS in facilitating cell-specific transcriptional programs.

Looking ahead, future research will investigate the mechanistic foundations of Egr2-AS-mediated regulation. This includes exploring the impact of transcription factor feedback mechanisms and the therapeutic potential of targeting antisense RNAs to enhance nerve repair. Additionally, we aim to examine the specific genomic regions where the antisense RNA interacts with various epigenetic regulators and identify the external factors that influence these interactions. By advancing our understanding of these complex regulatory processes, we can pave the way for innovative RNA-based therapies that could significantly improve outcomes in nerve injury and demyelinating conditions, ultimately transforming therapeutic strategies in the field.

## 5. Patents

Drs. Tapinos, Karambizi, and Martinez Moreno have filed a PCT WO/2023/215842 patent application regarding this work, entitled “Sustained release nucleic acid formulations for treatment of peripheral nerve demyelination”.

## Figures and Tables

**Figure 1 biomedicines-12-02594-f001:**
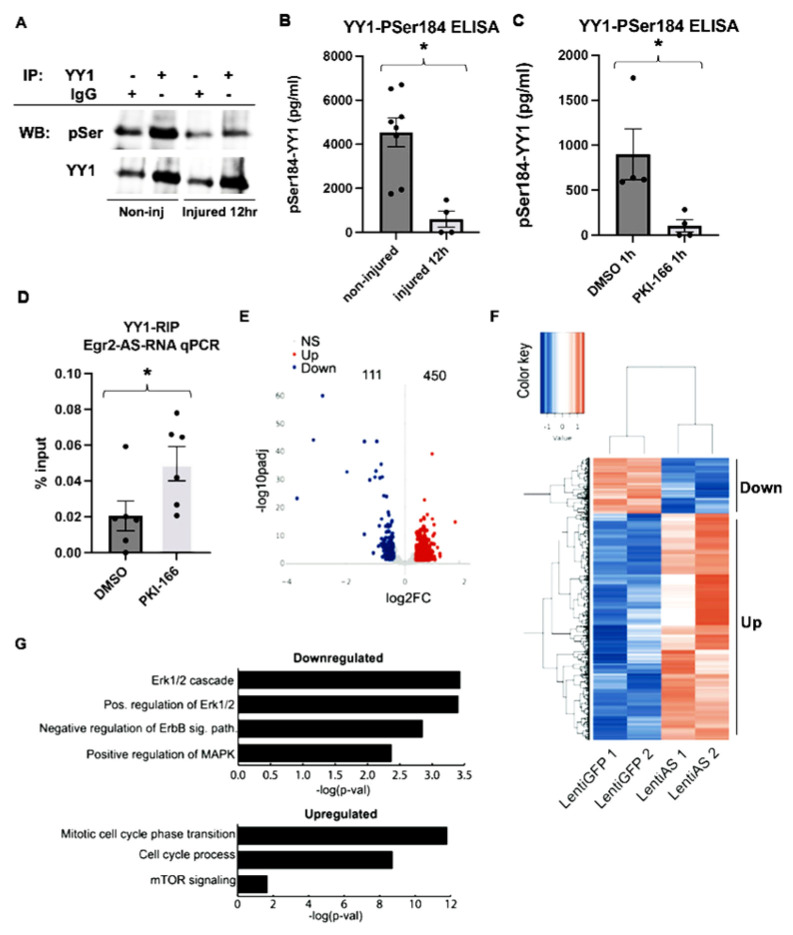
Expression of the EGR2-AS in SCs resulted in gene expression changes. (**A**) Western blot using total YY1 and phosphoserine antibodies following immunoprecipitation with a total YY1 antibody in non-injured sciatic nerves or 12 h after sciatic nerve transection. An isotype matched IgG was used as control. Sciatic nerve transection inhibits serine phosphorylation of YY1. (**B**) ELISA using our specific pSer184-YY1 antibody showed significant reduction in pSer184-YY1 12 h after sciatic nerve transection compared to contralateral uninjured nerves. Significance was calculated with a Student’s *t*-test (N = 4–8, * *p* < 0.005, dF = 10). (**C**) Inhibition of pErbB2-Y1248 with PKI-166 for 1 h resulted in significant inhibition of pSer184-YY1. Significance was calculated with a Student’s *t*-test (N = 4, * *p* < 0.05, dF = 6). (**D**) YY1 RIP followed by qPCR for the detection of the Egr2-AS shows increased binding to YY1 after inhibition of ErbB2 with PKI-166. Significance was calculated with a Student’s *t*-test (N = 4, * *p* < 0.05, dF = 4). (**E**) Volcano plot showing log-fold changes in gene expression following expression of the EGR2-AS in SCs. A total of 450 genes were significantly upregulated and 111 downregulated compared to control SCs. (**F**) Clustering of upregulated and downregulated genes in SCs expressing the EGR2-AS compared to control SCs (n = 2). (**G**) The downregulated genes were enriched for ERK1/2, ERBB, and MAPK-regulated biological processes, while the upregulated genes were enriched in cell cycle regulators and mTOR-regulated processes.

**Figure 2 biomedicines-12-02594-f002:**
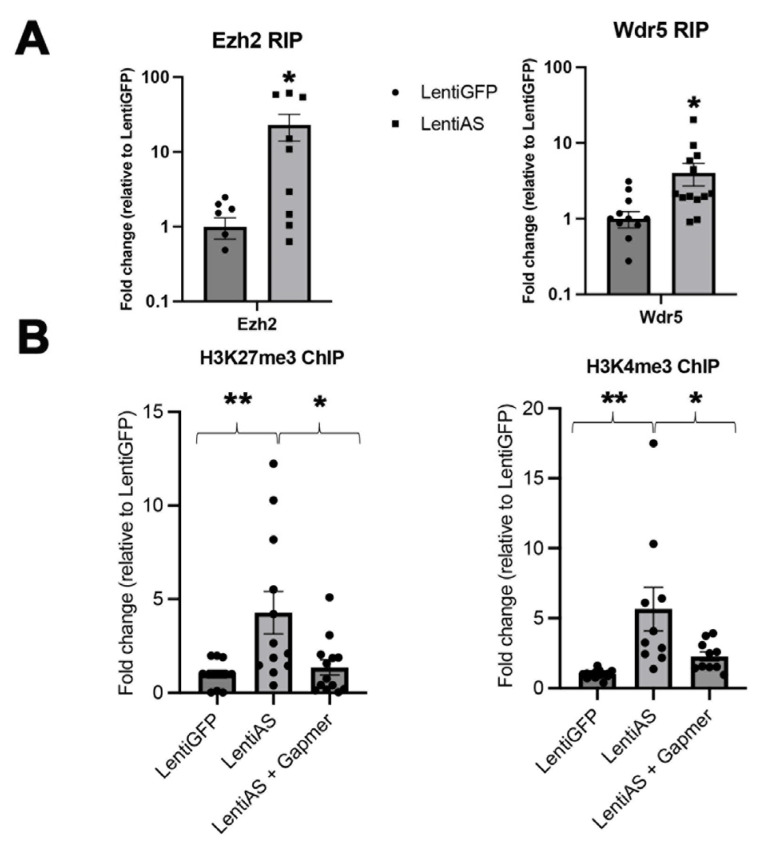
The EGR2-AS recruits EZH2 and WDR5 and enables targeting of H3K27me3 and H3K4me3 on *EGR2* and *C-JUN* promoters. (**A**) RIP experiments after the EGR2-AS expression in SCs, with antibodies against EZH2 and WDR5. Significance was calculated with a Student’s *t*-test (for the WDR5 RIP, N = 13, five biological replicates, * *p* = 0.040, dF = 27. For the EZH2 RIP, N = 9, three biological replicates, * *p* = 0.026, dF = 16). (**B**) ChIP experiments following expression of the EGR2-AS in SCs and its effect on H3K27me3 binding on *EGR2* promoter and H3K4me3 binding on C-JUN promoter, respectively. Incubation of cells with oligonucleotide GapmeRs against the EGR2-AS inhibits the AS-RNA-induced binding of H3K27me3 and H3K4me3 on the EGR2 and C-JUN promoters. For the H3K27me3 ChIP, N = 13, five biological replicates, and one technical replicate, * *p* = 0.020, dF = 23, ** *p* = 0.0094, dF = 22. For the H3K4me3 ChIP, N = 15, five biological replicates and one technical replicate, * *p* = 0.049, dF = 18, ** *p* = 0.0012, dF = 23.

**Figure 3 biomedicines-12-02594-f003:**
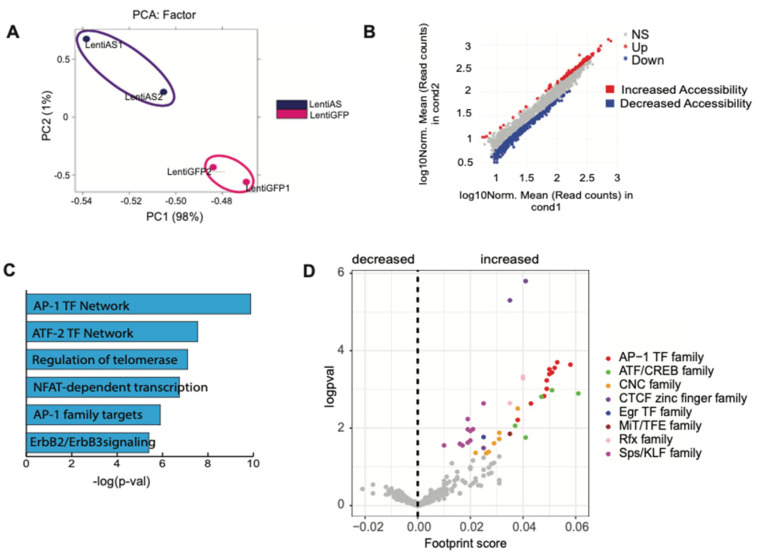
Expression of the EGR2-AS induces chromatin remodeling and increased binding of the AP-1/JUN TF family. (**A**) Sample correlation analysis based on ATAC-seq peak location and intensity. PCA result of all samples is plotted as a 2D graph with PC1 as *X*-axis and PC2 as *Y*-axis. Note that variable 1 is strong (98%) enough to divide into the two hierarchical groups. (**B**) Scatterplot comparing ATAC-seq signal intensities across all open chromatin sites between the Lenti-AS group compared to the Lenti-GFP group. Significant changes correspond to an FDR-adjusted *p* value below 0.05 and an absolute log2 fold change above 1.5. The diagonal is shown as a gray area and is a reference indicating regions with no change in chromatin accessibility. Colored dots indicate differences in accessibility. (**C**) GO analysis among genes located in the vicinity of regions with increased chromatin accessibility after the EGR2-AS overexpression. (**D**) Volcano plot of differential TF footprint score versus the significance (logpval) revealed that expression of the EGR2-AS induces significant increase in the footprint of AP-1, CTCF, ATF/CREB, CNC, EGR, MiT/FTF, Rfx, and KLF TF families.

**Figure 4 biomedicines-12-02594-f004:**
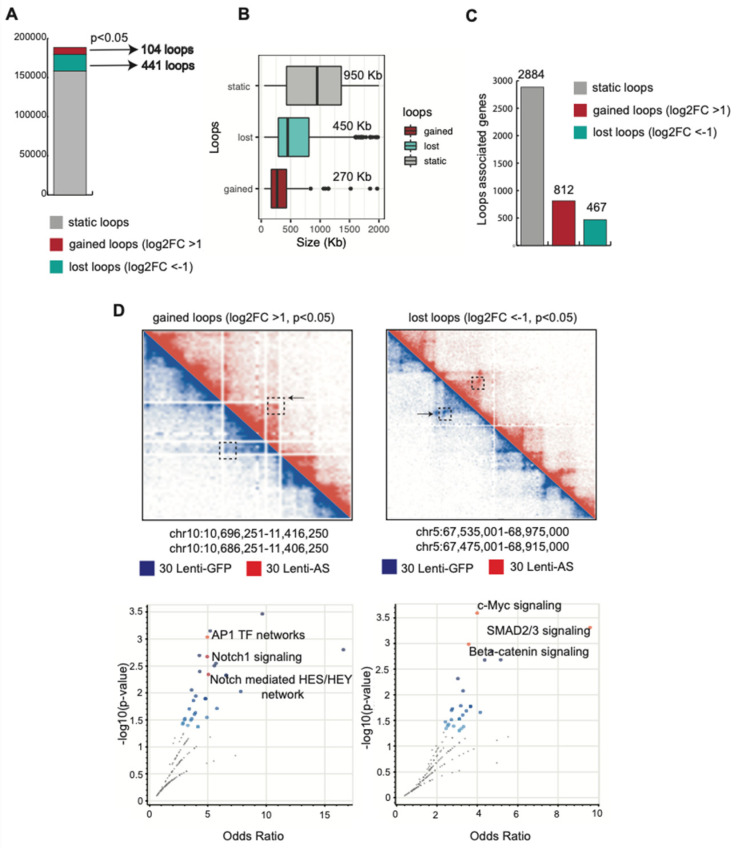
Hi-C following expression of the EGR2-AS in SCs shows 3D genome reorganization. (**A**–**C**) Histograms showing the significant total loops via the HiCDCPlus software version 1.0.0. Static loops (not gained or lost) are gray, lost loops are green, and gained loops are red (log2FC cutoff = 1, *p* < 0.05). (**D**) Detail of the Hi-C maps on the chromosomes 10 and 5 showing examples of gained and lost loops, respectively. The annotated loop anchors to the promoter of genes are represented by their main pathways (plotted as odds ratios by their corresponding *p* values).

**Figure 5 biomedicines-12-02594-f005:**
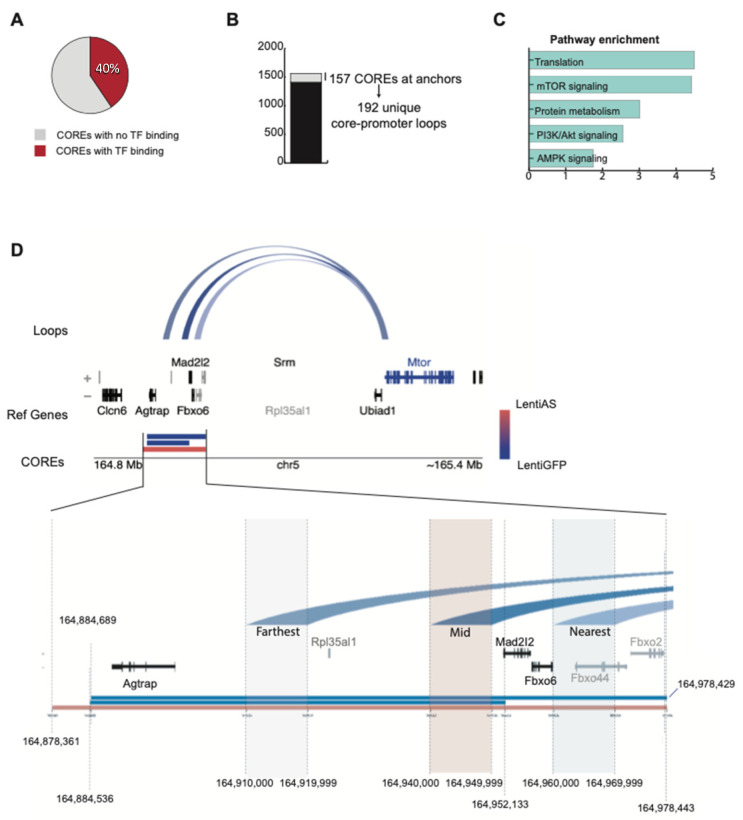
Reconstruction of long-range interactions between COREs and associated promoters. (**A**) Sub-setting of COREs based on transcription factor footprints, with 40% of COREs occupied by at least 1 TF footprint. (**B**) Hi-C revealed that 157 out of 1563 COREs form long distance interactions with promoters. (**C**) Pathway enrichment analysis of CORE-interacting genes. (**D**) Example of CORE–promoter interactions at the mTOR genomic region at chromosome 20. COREs are colored by samples: red and blue for the AS-RNA and GFP control, respectively. Reference genome is color-coded for expressed and non-expressed genes as black and grey, respectively.

**Figure 6 biomedicines-12-02594-f006:**
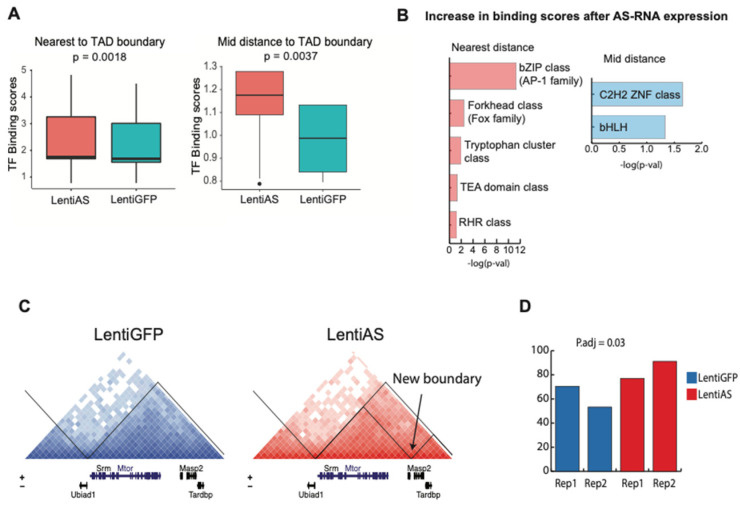
Expression of the EGR2-AS results in changes between the mTOR promoter and its cis-regulatory elements. (**A**) Change in TF binding scores of all TFs at nearest and mid-distance to TAD boundary loop anchors following expression of the EGR2-AS (farthest to TAD boundary loop anchor could not be analyzed due to very low occupancy). (**B**) TF families that experienced the greatest increase in binding scores following expression of the AS-RNA. (**C**) Depiction of changes at mTOR harboring TAD and formation of a new interdomain boundary. (**D**) Bar-plots of normalized gene expression of mTOR in control and AS-RNA expressing cells (*p*.adj = 0.03).

## Data Availability

Hi-C analysis scripts are available at https://github.com/hyeyeon-hwang/HiC-Analysis-of-Egr2-AS-RNA accessed on 14 April 2022. All datasets (RNA-seq, ATAC-seq, Hi-C) have been deposited to the GEO Omnibus and can be accessed through: https://www.ncbi.nlm.nih.gov/geo/query/acc.cgi?acc=GSE201627 accessed on 16 April 2023. The lentivirus construct (Lenti-AS) is deposited to Addgene and is available to order (#177737).

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
