# Peer review of "Role of the Egr2 Promoter Antisense RNA in Modulating the Schwann Cell Chromatin Landscape"

_biomedicines, 2024, doi:10.3390/biomedicines12112594_

Round 1

Reviewer 1 Report (Previous Reviewer 1)

Comments and Suggestions for Authors

The authors have satisfactorily addressed my concerns and comments.

Reviewer 2 Report (Previous Reviewer 2)

Comments and Suggestions for Authors

The authors have made significant improvements to the manuscript, addressing all prior comments effectively. The clarity and flow of the text have been greatly enhanced, particularly with clearer definitions of technical terms and improved structuring of the Results section. The methodological approach is now more transparent, with clearer explanations of confounder adjustments and gene expression analyses. The study’s impact is well-articulated. The Introduction and Discussion are concise and focused, while the figures are now more accessible with detailed captions. Overall, the manuscript is ready for publication and makes a valuable contribution to the field.

This manuscript is a resubmission of an earlier submission. The following is a list of the peer review reports and author responses from that submission.

Round 1

Reviewer 1 Report

Comments and Suggestions for Authors

The manuscript by Martinez Moreno is on the role of the Egr2 antisense RNA (Egr2-AS) in Schwann cells and conveys three important findings: 1. Egr2-AS action is dependent on neuregulin signaling as this signal regulates Egr2-AS phosphorylation-dependent binding to YY1. 2. Egr2-AS targets EZH2 and WDR5 complexes to the Egr2 and c-Jun promoters and changes the expression of these genes via histone modifications. 3. Egr2 modulates the overall chromatin landscape.

For the most part, these findings are clearly laid out and substantiated. My main concern is that the manuscript fails to explain how the action of Egr2-AS on the Egr2 and c-Jun promoters relates to its influence on the chromatin landscape. Is it upstream of the global chromatin remodeling, part of it or downstream of it? If upstream, how can the authors exclude that it is the altered Egr2/c-Jun levels that actually drive the remodeling? If it is part of the remodeling or a consequence of it, how do loops, TADs, longe-range and CORE-promoter interactions change for the two genes? Some additional information should be provided.

Similarly, it would be nice to hear the authors’ ideas why the Egr2-AS recruits EZH2 complexes to the EGR2 gene and WDR5 complexes to the c-Jun gene.

Other comments:

Line 73: “Methods for cell cultures, Transfections, qPCR, TF activity assays and Primer sequences are included in the Supplemental Information.” I suggest to move them to the manuscript proper.

Line 81: The chosen cut-off of 1.3 for RNA-seq (I suppose it is really 1.3 and not a log2-fold of 1.3) is quite unusual and fairly low. Please explain why this cut-off was chosen. How many differentially expressed genes would be left with the standard cut-off of 2 (log2-fold change of 1)? If my impression is correct that changes in gene expression are mild, please be explicit about it.

Line 297: “We developed an affinity purified pSer184 specific antibody by immunizing rabbits with the phospho-peptide KSGKK(pSer)YLGGGAGAC”. I would expect some validation of this novel tool.

Figure 1G, 5C: The pathway analyses (BTW what is it: GO, GSEA? I did not find that information in M&M) highlights a number of altered processes. Is this a selection or are these all the terms that popped up? If it is a selection, how was it done? If not the ones with the lowest p-value were chosen, an explanation should be provided and an extended list should be shown in the supplement.

Lines 359ff: This is too complicated. Please work your way through RIP and ChIP one after the other.

Figure 2: This figure just serves as an example, but my comment also applies to other figures. There are panels with bar graphs where single data points are shown (2A) and others where this is not the case (2B). I would suggest to show single data points in all bar graphs throughout the manuscript.

Figure 2C,D and lines 389ff: What are the 109 sites with Egr2-AS-dependent increased accessibility and the 467 with decreased accessibility? Are you talking about gene promoters only? This would make sense but is not clearly stated. If these are promoters, is it not a bit unusual to find enrichment of CTCF sites?

Lines 392ff: “Promoter of genes such as Jun, Junb, Jund, Fosb, Fos saw a significant increase in accessibility following Egr2-AS overexpression (Figures 3D & S2B).” This is shown in Figure S2B, but not obvious from Figure 3D.

Figure 3E,F: is missing.

Hi-C analysis is fairly sensitive and an n = 2 may be borderline. Could the authors provide some extra evidence for the quality/reproducibility of their data?

Figure 5A: is illegible

It needs an Institutional Review Board Statement at the end of the manuscript (ahead of references) as the Schwann cells used in the study were prepared from sciatic nerves of rat pups.

Reviewer 2 Report

Comments and Suggestions for Authors

The current manuscript entitled “Role of the Egr2 promoter antisense RNA in modulating 2 Schwann cell chromatin landscape” demonstrates interesting issues about the implication of Egr2 promoter in signaling pathways in SCs. The current manuscript is well written and highlights the importance of the presented study. However, there are some weaknesses that need to be addressed before this research article becomes acceptable for publication. For example, in introduction, the authors could add further information about the main aspects of their research in order to give more emphasis on its significance. Additionally, the content of the study should be extensively reviewed by the authors for grammatical, syntax errors as well as typos. Please describe and specify the RNA-seq experiments. I don’t understand if the authors have performed sequencing, or they analyzed available datasets. The wet lab procedures should be further discussed in the materials and methods section. In the same manner, ATAC-seq section should be also rephrased. It would be nice if the authors moved table2 into the supplementary material. The reagents used are not so important to mention in the main text. Results are well organized and easy to understand. At the discussion section the authors should add a summarizing paragraph describing the importance of their findings as well as the future directions.

Comments on the Quality of English Language

Minor editing of English language required.